# Effect of Fish Oil Parenteral Emulsion Supplementation on Inflammatory Parameters after Esophagectomy

**DOI:** 10.3390/nu16010040

**Published:** 2023-12-21

**Authors:** Ana Suárez-Lledó Grande, Josep M. Llop Talaveron, Elisabet Leiva Badosa, Leandre Farran Teixido, Mónica Miró Martín, Jordi Bas Minguet, Sergio Navarro Velázquez, Gloria Creus Costas, Nuria Virgili Casas, Mónica Fernández Álvarez, María B. Badía Tahull

**Affiliations:** 1Pharmacy Department, Bellvitge University Hospital, University of Barcelona—IDIBELL, 08907 L’Hospitalet de Llobregat, Spain; 2General Digestive Surgery Department, Bellvitge University Hospital, University of Barcelona—IDIBELL, 08907 L’Hospitalet de Llobregat, Spain; 3Immunology Department, Bellvitge University Hospital, University of Barcelona—IDIBELL, 08907 L’Hospitalet de Llobregat, Spain; 4Endocrinology and Nutrition Department, Bellvitge University Hospital, University of Barcelona—IDIBELL, 08907 L’Hospitalet de Llobregat, Spain

**Keywords:** immunomodulatory, inflammation, pharmaconutrient, omega-3 lipid emulsions, liver metabolism

## Abstract

(Background) Esophagectomy (EPG) presents high morbidity and mortality. Omega-3 fatty acids (ω-3FA) are a pharmaconutrient with benefits for postoperative morbidity. Studies of ω-3FA administered parenterally after esophagectomy are scarce. This study proposes to investigate the effect of combining fish oil lipid emulsions (LE) administered parenterally with enteral nutrition support. (Methods) Randomization was 1:1:1 in three groups: Group A received a LE mixture of 0.4 g/kg/day of fish oil and 0.4 g/kg/day of LCT/MCT 50:50, Group B received 0.8 g/kg/day of fish oil LE, and Group C received 0.8 g/kg/day of LCT/MCT 50:50. Variables were measured at recruitment time and day +1, +3, and +5. Inflammatory variables studied were Interlukin-6, C-reactive protein (CRP), tumoral necrosis factor-α (TNF-α), IL-10, IL-8 and CD25s. Safety, nutritional parameters and complications were analyzed. (Results) Administration of ω-3LE in the immediate postoperative period did not modulate the earlier inflammatory response. Statistically significant differences were found in IL-6 and CRP overall temporal evolution but were not found when studying the type of LE administered or in patients needing critical care. Administration of ω-3 resulted in safe and improved hypertriglyceridemia, depending on the dose. (Conclusions) ω-3FA has no impact on the early inflammatory postoperative response assessed for a short period but was safe. More studies for longer periods are needed.

## 1. Introduction

Esophageal cancer presents high mortality rates (about 500,000 cancer deaths each year), ranking sixth among all cancer-related deaths [1]. As reported in a recent review [2], the worldwide five-year overall survival rate ranges from 15% to 25% [3]. The esophageal cancer incidence increases with age, is more prevalent in men, and is associated with risk factors such as smoking habit, obesity, and alcoholism [4,5]. 

The main treatment for esophageal cancer is still esophagectomy (EPG), with about one in five patients requiring surgical resection. EPG is a major surgical intervention involving total or partial esophageal resection and restoration of gastrointestinal continuity [4]. EPG is associated with high rates of morbidity and mortality [4,6] and frequently presents postoperative complications such as anastomosis dehiscence, chylothorax, or even pneumonia [4,6,7]. In addition to these surgical complications, trauma from major surgeries like EPG activates the cytokine cascade, resulting in an alteration of the immune system and the development of a systemic inflammatory response. Different studies demonstrated an association between postoperative complications or mortality and postoperative inflammatory markers such as C-reactive protein (CRP), interleukin-6 (IL-6), or tumor necrosis factor-α (TNF-α) [8]. So, it is reasonable to hypothesize that postoperative complications could be minimized if inflammatory markers are reduced through anti-inflammatory mechanisms.

There are, however, controversial results in the literature, as some enteral pharmaconutrients showed effectiveness in the reduction of postoperative complications [6], and early enteral nutrition (EN) can be beneficial in improving immunocompetence, reducing infection rates, and maintaining intestinal functionality. Omega-3 fatty acids (ω-3FA) are one of the pharmaconutrients with beneficial effects on postoperative morbidity [9,10]. ω-3FA are composed of eicosapentaenoic acid (EPA) and docosahexaenoic acid (DHA), which are essential polyunsaturated fatty acids (PUFAs) derived from fish oil, and their administration can reduce proinflammatory cytokine production and regulate eicosanoid synthesis derived from arachidonic acid (AA), which are mediators of inflammation [8,9,10]. Also, DHA and EPA are associated with reduced proinflammatory transcription factor NF-κB activation. Furthermore, it is known that these PUFAs lead to the production of resolvins E and D, that have anti-inflammatory properties as they inhibit the migration of neutrophils and their infiltration into areas of inflammation and inhibit interleukin-1β (IL-1β) production [8]. 

One previous study evaluating the perioperative parenteral administration of ω-3FA in major abdominal surgery showed beneficial effects on patient outcomes [11]. Also, a previous review of 16 clinical trials assessing the effect of enteral immunonutrition with ω-3FA in gastrointestinal surgery patients demonstrated their benefits in reducing infectious postoperative complications and length of stay, particularly among individuals with preoperative malnutrition [12]. However, recent reviews evaluating the effect on the postoperative inflammatory response after perioperative ω-3FA administration (mainly enteral) in patients undergoing gastrointestinal surgery or EPG could not demonstrate any significant effect, although reduced inflammatory markers were observed [2,8]. In addition, focusing on patients who underwent EPG, they could not demonstrate any significant reduction in infectious complications or anastomotic leakage; however, it did not increase in-hospital mortality [2]. 

Studies on parenterally administered ω-3FA in esophagectomized patients are scarcely reported in the literature. A recent meta-analysis assessed the effect of ω-3FA-supplemented parenteral nutrition on inflammatory and immune function in postoperative patients with gastrointestinal malignancy. The analysis showed that early intervention with ω-3FA (maximum dose received was 0.2 g/kg/day) reduces the inflammatory reaction and improves the postoperative curative effect and the immune suppression induced by conventional parenteral nutrition (PN) or tumor [13].

This study proposes to investigate the effect of combining fish oil lipid emulsions (LE) administered parenterally with EN support within the framework of a randomized clinical trial. The main objective of this study is to establish whether the intravenous administration of ω-3FA fish oil LE for five days in esophagectomized patients is effective in reducing inflammation, measured as the serum concentration of IL-6, and to determine if 0.8 g/kg/day doses are more effective than 0.4 g/kg/day doses in the normalization of IL-6. Secondary objectives are to determine any differences regarding other inflammatory, hepatic, nutritional, and safety parameters, as well as differences in postoperative complications and mortality during hospitalization. 

## 2. Materials & Methods

This work contains the results of a prospective, single-center, randomized, double-blind study in patients diagnosed with esophageal cancer who underwent EPG by the Ivor-Lewis or McKeown techniques (EudraCT number 2016-004978-18; REEC protocol code FAR-NP-2017-01) [14]. Subsequently, patients received an LE via continuous infusion of ω-3FA or a 50% mixture of ω-6 long-chain triglycerides (LCT)/short-chain triglycerides (MCT), along with standard EN support for five days post-surgery [14] (Impact Neutre® 500 mL containing ω-3, ω-6, and arginine, excluding glutamine). As per usual clinical practice in our hospital, every patient with an indication for esophagectomy will be visited by a nutritionist who will make a first nutritional assessment and follow up on the tolerance of this enteral support. Impact Neutre is initiated at 21 mL/h and titrated to 1500 mL daily for 7 days. Impact Neutre has a caloric input of 144 kcal/100 mL.

Patients who meet the following inclusion criteria were enrolled: 18 years old or older, any gender and any race; those able to give their written informed consent (IC) for the study and having access to the digestive tract. The exclusion criteria were the following [14]: hypersensitivity type 1 or idiosyncratic reactions to any of the LE components; pregnant or breastfeeding women; plasmatic triglyceride (TRG) concentration > 3 mmol/L; chronic treatment with corticosteroids or immunosuppressants in the last month; HIV diagnosis; transplanted; hepatic impairment classified as Child-Pugh grade B or C. Every patient could voluntarily refuse to continue in the study and the following situations were also considered treatment withdrawal criteria: presentation of any adverse effect related to the treatment; any patient who presents a Clavien-Dindo grade IV complication (i.e., life-threatening complication requiring intensive care unit (ICU) management). 

The patients recruited were randomized 1:1:1 into three groups:

Group A received an LE mixture of 0.4 g/kg/day of fish oil and 0.4 g/kg/day of LCT/MCT 50:50, 

Group B received 0.8 g/kg/day of fish oil LE and,

Group C received an LE mixture of 0.8 g/kg/day of LCT/MCT 50:50.

This allocation was blind for all patients and health personnel. The allocation sequence (computer-generated random numbers) was generated by the pharmaceutical researcher, and it was be performed by randomized blocks permutated with the block size defined at random. In the first step, blocks of size 3 with possible sequences ABC, ACB, CAB, CBA, and blocks of size 6 were defined. In a second step, the clusters were randomized, and in a third step, the sequences were randomized within the blocks. This method ensures that the groups are balanced throughout the study and that the researcher cannot predict the sequence. The pharmaceutical researcher was responsible for patient assignment to the corresponding intervention group and also assigned a numbered patient identification code for each patient in the trial. The composition of the emulsion administered to each patient was known by the pharmaceutical researcher responsible for clinical trials, the pharmaceutical researcher who validated its conduct, and the personnel who conducted it. 

Each subject was followed up for a year post-surgical intervention. Figure 1 illustrates the diagram flowchart of this study [14]. 

The variables studied were the following [15,16,17,18,19,20,21]: ✓Serum concentrations of inflammatory parameters: IL-6, CRP, TNF-α, interleukin-10 (IL-10), interleukin-8 (IL-8) and soluble interleukin-2 receptor (sIL-2R/CD25s). ✓Postoperative complications: measured as suture dehiscence, chylothorax, pneumonia, and other respiratory tract infections.✓Safety: measured as hepatic impairment, TRG, and alterations in coagulation parameters.✓Nutritional parameters: albumin, prealbumin, and lymphocytes. ✓Mortality during hospitalization.

Variables were recorded pre-operatively at the time of recruitment and at days 0, +1, +3, +5 after randomization. Variables at day 0 inform about the basal inflammatory status of patients prior to esophagectomy and parenteral administration of LE.

The demographics and clinical and analytical data generated were collected and recorded in the Data Collection Logbook following good clinical practices. 

Cytokine values were determined by an Enzyme-Linked Immunosorbent Assay (ELISA). Samples were centrifuged at 700× *g* one hour post-extraction and were then aliquoted and stored frozen at −80 °C until analysis. 

Relative to statistical analyses, descriptive statistics were employed for all variable data, such as baseline values for inflammatory, nutritional, hepatic, and safety parameters, using frequency tables. For continuous variables, statistical descriptions were used (n, average, standard deviation, value range, and median), while grouped percentages were used for categorical variables.

The statistical analysis was carried out according to the following protocol and intention-to-treat, which included patients who required ICU care.

For the univariate approach, analyses of variance (ANOVA) using the Scheffe post hoc multiple-comparison test were performed to determine differences in the variables studied at randomization time and on days 0, 1, 3, and 5 post-intervention. For categorical variables, the chi-square test was used when necessary.

For the multivariate approach, a general linear model of repeated measures was used: MANCOVA models were performed for the different dependent variables (inflammatory and safety variables) with temporal measurements at days 0, 1, 3, and 5, and were adjusted for the different LEs administered. To establish temporal differences between dependent variables, the F statistic was used (the assumed sphericity when variances between different levels were equal and the Greenhouse-Geisser F statistic when variances were unequal). The Bonferroni test was used to compare temporal effects within subjects. Significance was established at *p* < 0.05. For all MANCOVA models, inter-subject contrast tests were performed to study the association of variable values measured at different times and with respect to the group variable (following the protocol). The coefficient B of determination was established at a significance level of *p* < 0.05. When no significance is observed between the groups, the variables studied will be transformed into natural (Napierian) logarithms for all times (at days 0, +1, +3, and +5). 

Data were processed with SPPS v 22.

This clinical trial and its posterior analysis were carried out following good clinical practices and with the approval of the ethics committee of the University Hospital of Bellvitge. 

## 3. Results

During the recruitment period, forty patients fulfilling the inclusion criteria were enrolled. Six patients were excluded because they did not undergo Ivor Lewis or McKeown surgery (exclusion criteria). In addition, three patients were lost after recruitment due to the SARS-CoV-2 pandemic and did not follow randomization, and another three were removed due to withdrawal of the lipid infusion following medical criteria.

Thus, after inclusion, 28 patients were randomized 1:1:1 into three groups. Four patients needed assistance in the ICU. Finally, 24 patients were analyzed following the protocol. See these results in Figure 2.

Data collected at baseline did not present any significant differences between groups (Table 1). Related to the characteristics and staging of the tumor, 22 patients presented adenocarcinoma and six presented squamous carcinoma. None of them had metastasis diagnosed when surgery was indicated. Only one patient was staged based on tumor size as T1, five were stratified as T2, fourteen as T3, and eight as T4. 

In the univariate analysis, no significant differences were observed for any variable analyzed taking into account the type of lipid administered and the different times studied. The results for the main variables are depicted in Table 2.

We performed a multivariant analysis (MANCOVA). As shown in Table 3, only IL6, CRP, and TRG showed significant differences in global temporal evolution. Maximum values were reached on day +1 for IL-6, day +3 for CRP, and day +5 for TRG. The results for IL-6, CRP, and TRG can be seen in Table 3. No differences were found in the temporal evolution of any variable when studying each group depending on the type of LE administered.

To compare differences in inflammatory parameters according to group at different times (0, +1, +3, and +5 days), an inter-subject contrast test based on coefficient B was performed. Results are shown in Table 4. Significance could be observed only in TRG variation on day +5 and a tendency on day +3 in favor of the group receiving 0.8 g/kg/day of ω-3.

In the intention-to-treat multivariant analysis (MANCOVA), the inclusion of critically ill patients showed statistically significant differences in the global evolution of inflammatory parameters, regardless of the lipids administered. Statistically significant differences were found in the temporal evolution of CRP values when critical patients were studied. CRP presents the highest value at day +3.

Concerning the other inflammatory parameters evaluated (TNF-α, IL-10, IL-8, and sl2R/CD25s), no statistically significant differences were observed, neither in global evolution nor according to groups. 

MANCOVA analysis for the variables IL6 and CRP transformed into natural logarithms was repeated. The statistical difference in global temporal evolution is maintained, with no significant differences being observed between the groups studied.

Regarding safety, measured by liver test function, TRGs, and alterations in coagulation and nutritional parameters, groups receiving ω-3 LE demonstrated it to be safe and to achieve nutrition goals. In fact, TRG levels from LE administration were reduced quicker in groups receiving 0.8 and 0.4 g/kg/day of ω-3, and especially lower levels were observed in the group receiving 0.8 g/kg/day of ω-3 at days 3 and 5 post-intervention, as shown in Table 4.

Complications were analyzed immediately post-surgery and during one year of follow-up, with the findings presented in Table 5. According to the group, no significant difference was observed in mechanical complications derived from surgery, like leakage or fistula and chylothorax, even when they presented mainly in the control group. No significant difference was observed in the incidence of respiratory tract infections between groups. Only two patients presented surgery site infections (SSI) (8.3%), which were resolved after short-term antibiotic therapy. The number of patients without complications was similar between groups: five patients in the group receiving 0.4 g/kg/day of ω-3 LE; six in the group receiving 0.8 g/kg/day of ω-3 LE; and seven in the control group, with no statistical differences between them (*p* = 0.091). No patient died during their hospitalization.

## 4. Discussion

EPG is an aggressive surgery that involves a rapid initial elevation of inflammatory parameters that quickly return to normal, as seen in this study. The administration of an LE with an immunomodulatory effect in the immediate postoperative period did not modulate this earlier inflammatory activity. Thus, we found statistically significant differences in the overall temporal evolution of IL-6 and CRP that were not found when studying the type of LE administered. This situation is confirmed even when critically ill patients are included. The administration of ω-3 is not only safe at all-time points administered but, in such a short period, it demonstrates its ability to improve hypertriglyceridemia depending on the administered dose, showing a greater improvement with the highest ω-3 dose.

It is important to highlight that, in this study, ω-3 was administered intravenously, affording some advantages compared with enteral administration. First of all, due to its iso-osmolarity, intravenous LE administration does not require a central line, thereby avoiding any extra-invasive treatment for the patient; in fact, LEs may have protective properties for the vascular wall. Moreover, in the patients considered for the study, the LE could be administered simultaneously with EN, so gastrointestinal tract stimulation was not interrupted and the caloric goal was achieved. Also, the use of only one parenteral pharmaconutrient (ω-3FA) and at a higher dose than in previous studies facilitates the caloric goal, achieves faster incorporation of ω-3FA into the cellular membranes and, consequently, into the inflammatory cascade [22,23], and enables the study of the immunomodulatory effect of one nutrient without any interference from other nutrients. When immunonutrition is administered enterally, the commercialized products available are combinations of different immunonutrients

Regarding inflammatory parameters, IL6 and CRP presented early high levels but returned to levels similar to baseline at day +5. Similar results were shown in a recent meta-analysis [13] that evaluated the evolution of inflammatory and immune function in postoperative patients with gastrointestinal malignancy after administration of enriched PN with ω3 LE. They also demonstrated that CRP levels show a rapid change in acute trauma and could reflect a change in the inflammatory response. Patients presented increased CRP levels within the first 4–12 h post-surgery, reaching the peak at 24–72 h, and returned to baseline on day +14. There were significant differences between groups on day +6, in favor of the ω3 group that presented lower levels than the control group, concluding that PUFAS can reduce the post-surgical inflammatory response in patients with gastrointestinal tumors. However, it is important to highlight that one of the limitations reported by the authors of this meta-analysis was that all studies included were carried out in Chinese populations and were also seeking a nutritional goal, so these results might not be extrapolatable to all populations. 

Previous studies evaluating the effect of PN enriched with ω3 LE on the immune response of patients with gastric or colorectal tumors found significant differences between groups in reducing the inflammatory response, measured by parameters like IL-6, CRP, and TNF-α [18,21]. In our previous study [24], we also observed that for longer periods of PN, inflammation in the presence of plasmatic phytosterols (PS) exhibits a synergistic effect on impairing hepatic function, mainly altering GGT but also ALT.

In this study, no significant differences were observed between groups for the other inflammatory markers: IL8, IL10, TNF-α, and CD25s. In a previous study that evaluated the effect of PN enriched with ω3 LE on reducing the levels of CD25s and procalcitonin (PCT) and improving cellular immunity, they demonstrated significant differences in PCT and cellular immunity; however, there were no differences between groups for CD25s [20]. Similar results were obtained in another study that assessed the influence of PN enriched with glutamine and/or ω3 on the inflammatory response of colorectal cancer patients measured as IL8 and cellular immunity. Cancer-induced alterations in cellular immunity could be corrected in the ω3-supplemented group, but there were no significant differences between groups in IL8 levels [19]. Furthermore, a recent meta-analysis [25] evaluated the influence of ω3 on humoral and cellular immunity and nutritional status in colorectal cancer patients. It demonstrated the effectiveness of ω3 administered postoperatively as enriched PN or EN in improving humoral and cellular immunity. They concluded that ω3 PUFAS can be a potential immunonutrition therapy in the post-operative care of patients with colorectal cancer. These studies suggest the need for further research to evaluate this anti-inflammatory effect. Taking this into account along with our study, IL8 and CD25s might not be suitable markers to evaluate inflammation in critical patients, which might be better evaluated using cellular immunity. 

Concerning morbimortality, there were more respiratory tract infections in the groups receiving 0.4 than in the control group and in the 0.8 g/kg/day of the ω-3 group (n = 2 vs. n = 1 and n = 1, respectively). Therefore, we could not demonstrate that ω3 LE could prevent infections as suggested in the hypothesis; however, these infections were not complicated and were easily resolved. A recent review^2^ analyzed the administration of ω3 as enriched EN in the perioperative period, demonstrating that enriched EN could reduce overall infectious complications. However, they also recognized that respiratory tract infections after EPG, which is one of the commonest complications, can be caused by other factors independent of the immunonutrition administered, e.g., surgical trauma, postoperative immunosuppression, or sputum accumulation. Even so, there were no significant differences in the incidence of pulmonary infection between the enteral immunonutrition group (EIN) and the EN group (Relative Risk = 0.96, Confidence Interval: 0.73–1.27, *p* = 0.79). They also suggested that it could be related to the surgical technique, it is well known that the pain of the surgical incision in EPG inhibits the patient’s voluntary cough, and impaired expectoration could have an impact on pulmonary infection. Related to SSI, only two patients presented this complication, which was resolved with antibiotic therapy. Every patient undergoing esophagectomy in our hospital receives a single dose of metronidazole 1 g intravenously and two single doses of cefuroxime 1.5 g intravenously. The incidence of SSI in this study was similar to that of a recent review [26], where patients who received a single dose of first-generation cephalosporins and nitroimidazole had in a 9.7% SSI rate. They concluded that prophylaxis with antibiotics can reduce SSI in upper gastrointestinal surgery, like esophagectomy, and also that a single-dose regimen is non-inferior to multiple doses, which is preferred to reduce microbiologic resistance.

Finally, the intravenous administration of ω3 LE for five days after surgery was safe, as well as being associated with lower TRG levels compared with the control group, especially in the 0.8 g/kg/day group. This suggests that administration of ω3 LE for a short period can control lipid metabolism and also does not have an impact on liver test functions (GGT, FA, ALT, and AST). Safety is warranted as ω-3 LE administration has demonstrated no liver damage compared with LCT LE, as ω-3 LEs do not contain phytosterols, which are involved in hepatotoxicity.

One limitation of this study is the small number of patients included; however, they are uniformly distributed between groups, and the results obtained remain consistent across the different statistical tests carried out. Further studies with a greater number of patients could provide more results; they could also analyze the influence of tumor staging on the inflammatory response and the influence of cellular immunity as a possible marker of inflammation. 

## 5. Conclusions 

In this study, we found that ω-3 FA has no impact on the control of the early inflammatory postoperative response assessed over a short period, so it cannot be ruled out that the administration of fish oil LE could improve this inflammatory response in complex situations, like those characterized by major metabolic stress and prolonged in time. Administration of fish oil proved to be safe and, even for a short period, could normalize TRG levels, proving to be dose-dependent and able to control lipid metabolism. Further studies and longer administration periods are needed to evaluate the utility of fish oil as a pharmaconutrient in controlling inflammatory processes and the influence of nutritional assessment and enteral immunonutrition on this response. Moreover, further studies are needed to evaluate the inflammatory response measured by cellular immunity.

## Figures and Tables

**Figure 1 nutrients-16-00040-f001:**
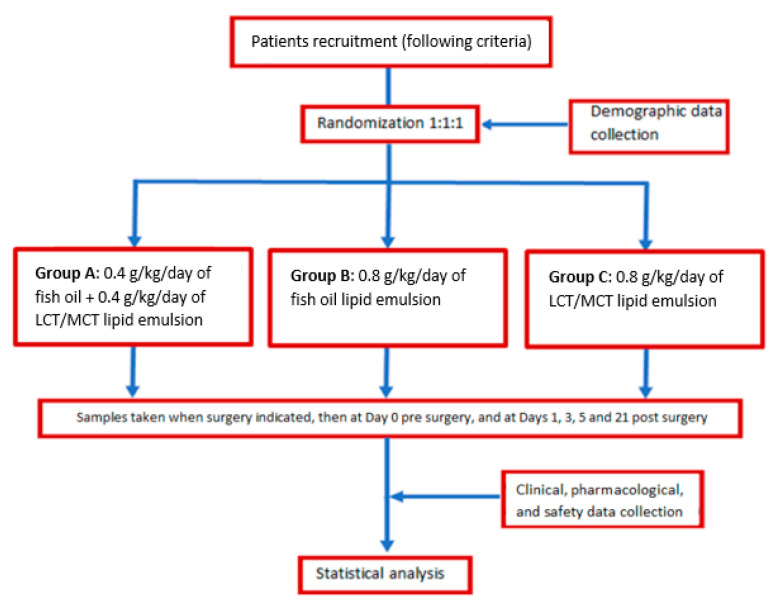
Study design flow chart.

**Figure 2 nutrients-16-00040-f002:**
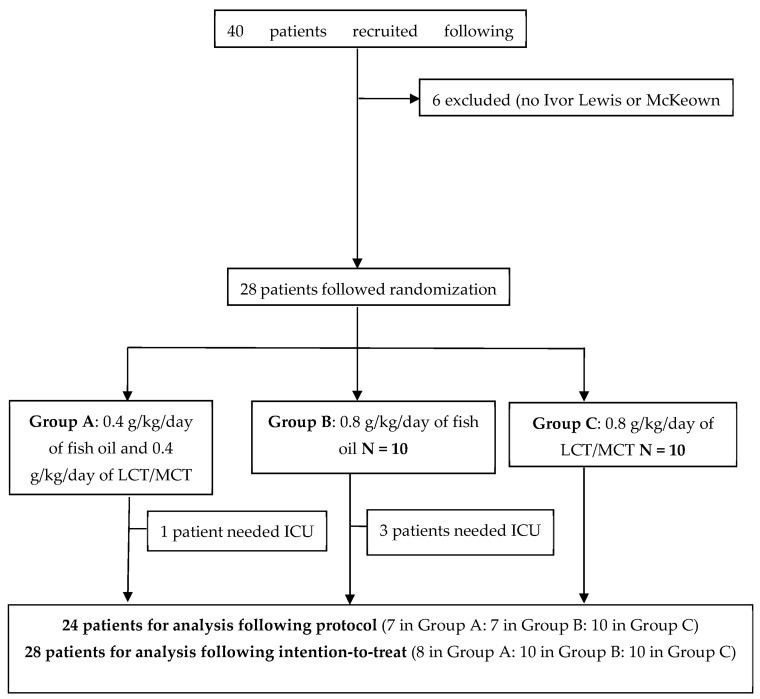
Flowchart of patients randomized in the study and follow-up.

**Table 1 nutrients-16-00040-t001:** Baseline data of all patients included in the analysis by intention-to-treat.

Parameter	Group 0.4 g/kg/day of ω-3 + 0.4 g/kg/day of LCT/MCT (N = 8)	Group 0.8 g/kg/day of ω-3 (N = 10)	Group 0.8 g/kg/day of LCT/MCT (N = 10)	Significance
Demographics	Mean (SD)	Mean (SD)	Mean (SD)	*p* *
Age, y	62.88 (8.47)	58.60 (10.46)	63.2 (7.77)	*p* > 0.05
Weigh, kg	69.40 (11.25)	74.45 (20.17)	76.05 (9.74)	*p* > 0.05
BMI	25.41 (4.30)	26.21 (6.71)	26.10 (2.87)	*p* > 0.05
Nutritional				
Albumin, g/L	43.29 (2.06)	45.40 (2.46)	45.80 (2.97)	*p* > 0.05
Prealbumin, mg/L	247.88 (31.42)	264.10 (38.32)	275.00 (31.06)	*p* > 0.05
Lymphocytes, ×10^9^/L	1.25 (0.56)	1.31 (0.59)	1.58 (0.71)	*p* > 0.05
Liver function and safety				
Total Bilirubin, μmol/L	7.25 (3.99)	6.60 (2.32)	5.80 (2.20)	*p* > 0.05
Alanine aminotransferase, μkat/L	0.42 (0.29)	0.37 (0.12)	0.39 (0.12)	*p* > 0.05
Aspartate aminotransferase, μkat/L	0.41 (0.15)	0.35 (0.08)	0.37 (0.08)	*p* > 0.05
γ-Glutamyl transferase, μkat/L	1.87 (1.62)	0.89 (0.77)	0.76 (0.48)	*p* > 0.05
Alkaline phosphatase, μkat/L	1.38 (0.64)	1.23 (0.59)	1.17 (0.29)	*p* > 0.05
Triglycerides, µmol/L	1.63 (0.67)	1.32 (0.64)	1.57 (0.52)	*p* > 0.05
INR	1.00 (0.07)	0.99 (0.09)	0.99 (0.08)	*p* > 0.05
Prothrombin time	1.00 (0.07)	0.97 (0.03)	0.96 (0.04)	*p* > 0.05
Platelets, ×10^9^/L	269 (107.14)	217.70 (52.44)	251.40 (49.22)	*p* > 0.05
Inflammatory				
Interleukin-6, pg/mL	3.54 (4.05)	3.24 (4.29)	1.73 (3.00)	*p* > 0.05
C-Reactive protein, mg/L	5.64 (5.13)	5.85 (8.88)	5.82 (7.98)	*p* > 0.05
Tumor necrosis factor-α, pg/mL	6.84 (7.04)	15.12 (10.09)	15.10 (11.59)	*p* > 0.05
Interleukin-10, pg/mL	0.00 (0.00)	3.53 (9.66)	1.75 (3.73)	*p* > 0.05
Interleukin-8, pg/mL	0.39 (1.12)	0.00 (0.00)	0.11 (0.36)	*p* > 0.05
Soluble receptor CD25, pg/mL	1.14 (2.19)	22.00 (46.39)	5.14 (16.25)	*p* > 0.05

SD, standard deviation. * Significance is defined as *p* < 0.05 in the ANOVA analysis. BMI, body mass index. INR, international normalized ratio.

**Table 2 nutrients-16-00040-t002:** Principal parameters studied (IL-6, CRP, and triglycerides) among groups at the different times studied.

Variable		N	Mean	SD	F	Significance
IL6 PRE	LCT/MCT 0.8 g/kg/day	4	2.6650	3.22215	0.398	0.680
	ω3 0.4 g/kg/day	6	4.7300	7.22580		
	ω3 0.8 g/k g/d	5	2.1540	2.01955		
IL6-0	LCT/MCT 0.8 g/kg/day	10	1.7330	3.00331	0.425	0.659
	ω3 0.4 g/kg/day	7	3.4971	4.37432		
	ω3 0.8 g/kg/day	7	2.6129	4.53815		
IL6-1	LCT/MCT 0.8 g/kg/day	10	195.530	153.4789	1.520	0.242
	ω3 0.4 g/kg/day	7	115.614	100.8809		
	ω3 0.8 g/kg/day	7	267.486	218.0746		
IL6-3	LCT/MCT 0.8 g/kg/day	9	83.6078	70.97293	0.412	0.668
	ω3 0.4 g/kg/day	7	62.4571	46.59213		
	ω3 0.8 g/kg/day	7	59.5143	50.76322		
IL6-5	LCT/MCT 0.8 g/kg/day	9	33.7489	24.16470	0.431	0.656
	ω3 0.4 g/kg/day	7	37.1571	6.18543		
	ω3 0.8 g/kg/day	7	28.3543	15.93263		
PCR PRE	LCT/MCT 0.8 g/kg/day	3	5.100	1.6093	1.653	0.240
	ω3 0.4 g/kg/day	6	28.133	25.3114		
	ω3 0.8 g/kg/day	4	11.000	15.8108		
PCR-0	LCT/MCT 0.8 g/kg/day	10	5.820	7.9802	0.147	0.864
	ω3 0.4 g/kg/day	7	5.214	5.3899		
	ω3 0.8 g/k g/d	7	7.471	10.3147		
PCR-1	LCT/MCT 0.8 g/k g/d	9	65.144	25.1391	1.108	0.350
	ω3 0.4 g/k g/d	7	111.343	102.8056		
	ω3 0.8 g/k g/d	7	92.543	40.0408		
PCR-3	LCT/MCT 0.8 g/k g/d	8	163.125	88.0023	0.782	0.472
	ω3 0.4 g/kg/day	6	143.967	44.8463		
	ω3 0.8 g/kg/day	7	192.971	68.3032		
PCR-5	LCT/MCT 0.8 g/kg/day	10	96.950	51.2645	1.315	0.290
	ω3 0.4 g/kg/day	7	145.029	56.6069		
	ω3 0.8 g/kg/day	7	122.829	75.9096		
TRG PRE	LCT/MCT 0.8 g/kg/day	2	1.3600	0.55154	0.403	0.681
	ω3 0.4 g/kg/day	6	1.6750	0.52630		
	ω3 0.8 g/kg/day	3	1.4100	0.51971		
TRG-0	LCT/MCT 0.8 g/kg/day	10	1.5660	0.51810	0.365	0.699
	ω3 0.4 g/kg/day	7	1.6714	0.71539		
	ω3 0.8 g/kg/day	7	1.3771	0.76661		
TRG-1	LCT/MCT 0.8 g/kg/day	9	1.2578	0.38803	0.417	0.665
	ω3 0.4 g/kg/day	6	1.5033	0.51949		
	ω3 0.8 g/kg/day	7	1.2729	0.72318		
TRG-3	LCT/MCT 0.8 g/kg/day	9	1.5522	0.49068	0.820	0.455
	ω3 0.4 g/k g/d	6	1.3183	0.37333		
	ω3 0.8 g/k g/d	7	1.2557	0.56806		
TRG-5	LCT/MCT 0.8 g/kg/day	10	1.9030	0.51968	2.338	0.121
	ω3 0.4 g/kg/day	7	1.7786	0.56708		
	ω3 0.8 g/kg/day	7	1.3614	0.46208		

PRE: values measured at randomization time. 0: values measured on intervention day. 1, 3, 5: values measured at days 1, 3, and 5 post-intervention. TRG: triglycerides. SD: standard deviation. F: Snedecor’s F. Significance is defined as *p* < 0.05 in the univariate analysis following protocol.

**Table 3 nutrients-16-00040-t003:** Values of IL6, CRP and TRG measured at different times and their global temporal evolution of variables analyzed.

	IL6		CRP		TRG	
Time	Mean	SD	Mean	SD	Mean	SD
Day 0	1411	1025	3657	1512	1669	0.187
Day +1	235,288	51,374	60,100	7031	1191	0.126
Day +3	92,988	24,627	177,743	31,715	1605	0.176
Day +5	32,293	8983	95,100	20,074	1879	0.190
Global temporal evolution	<0.001 *		<0.001 *			0.035 *

SD: standard deviation. * F statistic: Greenhouse-Geisser and assumed sphericity. Multivariate analysis following protocol (MANCOVA).

**Table 4 nutrients-16-00040-t004:** Comparisons by day between groups and the control group for IL6, CRP, and TRG values adjusted by the linear model based on coefficient B.

		IL6		CRP		TRG	
Time	Group	B	Significance	B	Significance	B	Significance
Day 0	ω3 0.8 g/kg/day vs. group C	1828	0.326	2193	0.522	−0.317	0.323
	ω3 0.4 g/kg/day vs. group C	2132	0.279	2414	0.516	−0.024	0.942
Day +1	ω3 0.8 g/kg/day vs. group C	41,553	0.620	20,330	0.179	0.008	0.978
	ω3 0.4 g/kg/day vs. group C	−71,625	0.419	11,643	0.471	0.176	0.545
Day +3	ω3 0.8 g/kg/day vs. group C	22,472	0.653	19,307	0.603	−0.432	0.092
	ω3 0.4 g/kg/day vs. group C	−13,712	0.794	−7986	0.843	−0.408	0.132
Day +5	ω3 0.8 g/kg/day vs. group C	49,085	0.292	48,850	0.152	−0.664	0.022
	ω3 0.4 g/kg/day vs. group C	16,220	0.738	59,900	0.107	−0.273	0.353

0: values measured on the same day of the intervention. 1, 3, 5: values measured at days 1, 3, and 5 post-intervention. TRG: triglycerides. B: coefficient of determination. Significance is defined as *p* < 0.05.

**Table 5 nutrients-16-00040-t005:** Complications observed in the different groups following protocol.

	ω 3 0.4 g/kg/day N = 7	ω 3 0.8 g/kg/day N = 7	Control N = 10	Significance
Leakage/Fistula	0	0	3	0.091
Sepsis	0	0	0	-
Pneumonia	0	0	0	-
Chylothorax	0	0	2	0.217
Respiratory tract infections	2	1	1	0.482

Significance is defined as *p* < 0.05.

## Data Availability

The data supporting the findings of this study are available from the corresponding author upon reasonable request. The data are not publicly available due to privacy.

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
