# Peer review of "Effect of Fish Oil Parenteral Emulsion Supplementation on Inflammatory Parameters after Esophagectomy"

_nutrients, 2023, doi:10.3390/nu16010040_

Round 1

Reviewer 1 Report

Comments and Suggestions for Authors

The article proposes a randomized clinical trial aiming to investigate the effects of intravenous administration of ω-3FA fish oil LE on inflammation in patients undergoing esophagectomy for cancer.

The study appears well-designed, showing clear results with a good novelty but showing limited significance of conclusions. 

The introduction section provides sufficient background and useful references to introduce the body of the paper. 

The section “material and methods” is well organized, the number of patients in the three groups is quite uniform but still small for the validity of the study's conclusions.

Results section appears well reasoned, with a well performed statistical analysis. Figures and tables are well structured and detailed, and clearly show the data. I would recommend that the staging and histotype of the oesophageal tumour also be introduced among the table data and possibly study any correlations with the inflammatory response. Among nutritional and inflammatory parameters, I would also recommend adding transferrin, white blood cells count as well as procalcitonin and counts of CD4/CD8, IgG, IgM and IgA levels.

The discussion section could be better structured. It would be interesting to know more details about antibiotic therapy and additional complications, such as SSI infections. An interesting systematic review about SSI and antimicrobial prophylaxis in esophagogastric surgery may enrich your discussion (doi: 10.3390/antibiotics11020230). It would be not secondary to know whether the patients included in the study have undergone nutritional screening and prehabilitation protocols.

Conclusions are rather meager. Future goal might be to investigate how prehabilitation might also affect postoperative inflammation, along with parenteral -3FA supplementation.

Comments on the Quality of English Language

English language requires a minor revision in all document.

Author Response

Please find attached the answer to your comments. Thank you so much

Reviewer 2 Report

Comments and Suggestions for Authors

Very interesting study. However I have several questions? 

1.  Is this study registered officially? (ex. clinicaltrials.gov)

 If this is registered, document it.

2. How to randomize the group? randominzation method was not documented.

3. Why are you give the immunonurients for EN?

  Enteral w-3 can affect the inflammatory response or clinical outcomes. 

 To compare the effects of parenteral w-3, enteral w-3 should be limited. 

4. How many lipid were delivered including enteral & parenteral?

   How many calories were delivered?

Author Response

(The authors gave the same response as above.)

Round 2

Reviewer 2 Report

Comments and Suggestions for Authors

well documented to the question